# Clinical and Radiographic Outcomes of Sloped-Shoulder Implants in the Posterior Mandible: A Retrospective Study

**DOI:** 10.3390/dj13100466

**Published:** 2025-10-11

**Authors:** Guillem Esteve-Pardo, Javier Amigó-Bardají, Lino Esteve-Colomina

**Affiliations:** 1Clínica Dental Esteve SL, Group Aula Dental Avanzada, 03001 Alicante, Spain; 2Clínica Dental Amigó, Group Aula Dental Avanzada, 46001 Valencia, Spain

**Keywords:** sloped-shoulder implant, marginal bone loss, posterior mandible, implant survival, retrospective study, guided bone regeneration, cone beam computed tomography

## Abstract

**Background/Objectives:** This retrospective study aimed to evaluate the survival and marginal bone loss (MBL) of sloped-shoulder implants placed in the posterior mandible, and to explore the influence of both patient- and implant-related factors. **Materials and Methods:** All patients treated with sloped-shoulder-profile implants (Astra Tech Implant System, Dentsply Sirona, Bensheim, Germany) in the posterior mandible between 2012 and 2023 at two private clinics were included. Implant survival was analyzed with Kaplan–Meier estimates. MBL was measured from prosthesis delivery (baseline radiograph) to the most recent available radiograph. Outcomes were compared across thresholds of 0, 0.5, and 1.5 mm, which were considered radiographic success criteria. According to the 2017 World Workshop, peri-implantitis was not diagnosed solely based on MBL. Associations with potential risk factors (periodontitis, bruxism, and smoking) were explored. The study was approved by a local ethics committee (PI 106/2023); informed consent was waived due to the retrospective design and anonymization of data. **Results:** A total of 43 patients with 48 implants were included, with a mean follow-up of 40.1 months. The cumulative survival rate was 93.7%, with all failures occurring before 24 months. Mean MBL at the mesial and distal aspects was 0.27 mm and 0.39 mm, respectively. In 82.2% of implants, MBL remained ≤0.5 mm at a mean follow-up of 44.2 months. No statistically significant associations were found between risk factors such as periodontitis, bruxism, or smoking and implant outcomes. **Conclusions:** Sloped-shoulder implants in the posterior mandible showed high survival and stable marginal bone levels over the medium term. Their design may simplify treatment in oblique ridges, potentially reducing the need for GBR procedures.

## 1. Introduction

A collapse of the alveolar bone is an inevitable consequence of tooth extraction [1]. Resorption typically occurs in the buccal plate and is characterized by a combined vertical–horizontal reduction, resulting in an oblique ridge with an approximate height difference of 2 mm between the buccal and lingual bone plates [2].

In 2011, a specific implant was designed with a sloped platform configuration with the objective of achieving a superior adaptation to the moderately atrophied healed ridge (Figure 1). In such cases, conventional flat-platform implants necessitate positional adjustments, either by deeper placement on the lingual crest—which may result in increased bone resorption and/or greater probing depths [3,4], or by performing bone augmentation on the buccal side, which entails additional costs and prolongs the treatment period [5].

Pre-clinical experimental studies have demonstrated a reduction in marginal bone loss in comparison with the standard flat-platform implants [6,7]. Early clinical studies have demonstrated that the inclined shoulder of the implant facilitates prosthetically driven positioning by adapting to the oblique alveolar ridge, thereby reducing the need for guided bone regeneration or compensatory adjustments [8].

Management of oblique or resorbed ridges often requires guided bone regeneration (GBR), including horizontal or vertical augmentation, particulate grafts, onlay blocks, and/or the use of membranes. These procedures increase treatment complexity and morbidity. Sloped-shoulder implants were designed to adapt to the angulated crest, potentially reducing the need for GBR in moderately resorbed ridges.

From a clinical perspective, the introduction of the sloped platform design helps to circumvent the necessity for additional procedures, enabling a more natural alignment of the implant with the oblique alveolar ridge while preserving lingual bone and simplifying the entire procedure. A brief clinical follow-up of these implants in the posterior mandible demonstrated the maintenance of peri-implant keratinized tissue and bone level [9]. Furthermore, a three-year clinical study reported that both hard and soft tissues remained stable around these implants [10]. A recent randomized controlled trial (RCT) comparing conventional versus sloped platform implants found a reduction in surgical intervention time, a decrease in postoperative discomfort, and a satisfactory aesthetic result with the latter [11].

This type of implant has been employed for immediate placement in cases where there is a greater likelihood of buccal bone volume loss [12]. Additionally, sloped implants have been used in the two distal positions of the all-on-four concept, enabling the platform to be leveled with the bone when implants are placed in a tilted position [13,14]. A finite element analysis revealed that this resulted in a reduction in stress levels in comparison to the standard shoulder design [15].

The selection of an appropriate implant design is of critical importance in the posterior mandible, where bone quality and quantity are often limited. While functional load is significant in the posterior quadrants, poor mandibular bone quality can jeopardize implant stability and load transfer to the bone [16,17]. The sloped platform implant is particularly well-suited to cases of moderate resorption, offering significant advantages in prosthetic planning and clinical practice. By modifying the implant platform to align with the residual anatomy, it is possible to enhance the final implant position while maintaining stability and aesthetic outcomes.

In this retrospective study, the Profile implant was used to restore missing teeth in the posterior mandible in positions 5 to 7. A cohort of forty-eight implants in 43 patients was followed over a period up to ten years, with a medium of 44.2 months. Clinical results and peri-implant tissue changes are presented and discussed. The aims are to evaluate the survival rate and the radiographic marginal bone loss of sloped platform implants in the posterior mandible and to identify factors predictive of these outcomes.

## 2. Materials and Methods

### 2.1. Study Design/Sample

This retrospective study included all patients who received a sloped-shoulder implant in the posterior mandible between June 2012 and December 2023 at two private clinics with comparable characteristics. Only cases with available clinical records and at least one follow-up radiograph were included. No patients were excluded on the basis of systemic conditions, smoking, or history of periodontal disease, provided that a sloped implant was placed distal to the mandibular second premolar.

All implants corresponded to the Profile design in two versions: OsseoSpeed TX and OsseoSpeed EV (Astra Tech Implant System, Dentsply Sirona, Bensheim, Germany). Clinical, medical, and radiographic information from each patient was retrieved from the corresponding files to ensure a comprehensive dataset.

The study was conducted in accordance with the STROBE guidelines and the Declaration of Helsinki. Ethical approval was obtained from the Ethics Committee for Research with Medical Products of the Hospital of Elche (Alicante, Spain) in December 2023 (protocol PI 106/2023, on 27 December 2023). Given the retrospective nature of the investigation and the use of anonymized data from routine clinical care, the Committee explicitly waived the requirement for individual informed consent.

The primary outcome was the survival rates of the Profile implants placed in the follow-up period. The secondary outcome was the marginal bone loss (MBL) obtained with the most recent radiographic record.

### 2.2. Study Variables

The following clinical and radiographic data were recorded per implant:Age of the patient at the time of implant placement.Sex.Medical data.Presence or not of periodontal disease.Presence or not of bruxism.Date of implant placement.Location.Implant system and its measurements.Type of prosthetic restoration according to retention type.Type of abutment.Follow-up time with X-rays.Measurement in millimeters (mm) of MBL in mesial and distal made through 2D radiograph.Follow-up time with 3D X-ray when available.Measurement in mm of MBL in buccal and lingual performed by 3D X-ray.Occurrence of mechanical complications.Signs of inflammation, suppuration or bleeding of soft tissues.Early failures and time in months from implant placement.Late failures and time in months from implant placement.

### 2.3. Definitions

An implant is deemed a survivor when it exhibits MBL, yet remains non-mobile and provides prosthetic support without complications, whether biological or technical, and with complete patient satisfaction.

The failures were classified according to whether they occurred at an early or late stage. An early failure was defined as the loss of the implant prior to the placement of the prosthesis. Implant loss occurring after prosthesis placement, but more than six months after the latter, was defined as late failure.

In order to make a clinical diagnosis of bruxism in each patient, we have employed the Clinical Based Assessment (CBA) recently described by Manfredini and colleagues. This assessment explores three domains: joints and muscles, intraoral and extraoral tissues, and teeth and restorations [18].

Periodontal disease was diagnosed in accordance with the criteria set forth by the European Federation of Periodontology (EFP), which includes the presence of radiographic bone loss or probing depths of less than 4 mm, in addition to the occurrence of bleeding on probing (BoP) or pockets deeper than 6 mm [19].

Peri-implantitis was not diagnosed solely based on marginal bone loss. According to the 2017 World Workshop on Periodontology, peri-implantitis is defined as the presence of bleeding and/or suppuration on probing, increased probing depth (≥6 mm), and radiographic bone loss ≥ 3 mm compared with the expected initial remodeling. These criteria were not systematically assessed in this retrospective study; therefore, MBL > 0.5 mm was not interpreted as peri-implantitis [20].

The MBL inscription in millimeters was rounded to one decimal place, with the second decimal being rounded down if less than 0.5 or up if greater.

### 2.4. Clinical Procedure

The sloped implants were placed in inclined residual ridges at a minimum of six months following tooth extraction. All patients received a baseline radiograph immediately after prosthesis placement. This radiograph was considered the reference baseline for subsequent MBL evaluation.

The procedure employed for implant placement was that recommended by the manufacturer. Implants were positioned flush with the buccal cortical bone and taking into account the height of the lingual wall, which was typically 1.5 mm taller than the buccal.

All implants were treated with a non-submerged technique by placing a healing abutment of more than 2 mm, except for one case intentionally placed with a submerged approach. This isolated case was not considered as a variable in the survival or MBL analyses. The second stage was performed at least three months after placement. All prosthetic restorations were performed using titanium abutments. Atlantis abutments (Dentsply Sirona, Mölndal, Sweden) were used in single-unit sites and MultiBase (Dentsply Sirona, Mölndal, Sweden) or Uni Abutments in partial fixed prostheses, except for one patient who received a three-unit bridge cemented to Atlantis abutments. All crowns were screwed in, except in cases where cementing was necessary for various reasons (Figure 2 and Figure 3).

### 2.5. Radiographic Examination

The radiographic examination was conducted using the most current radiographs available for each patient at the time of the investigation.

Periapical X-rays were taken using a parallel technique and examined in order to obtain interproximal MBL data. Cone beam computed tomography (CBCT) X-rays, when available, were analyzed to obtain lingual and buccal measurements.

Measurements were conducted by three reviewers: the two operators and an independent examiner who was unaware of the patients. In cases where there were significant discrepancies, the evaluation was repeated until a consensus was reached.

All measurements were performed with the Sidexis 4 software (Sirona Dental Systems GmbH, Bensheim, Germany). In the analysis of the interproximal bone, the radiograph taken immediately after prosthesis delivery was considered the baseline. This was compared with the most recent available radiograph, allowing differentiation between initial bone remodeling after loading and progressive marginal bone loss during follow-up. In both analyses, 2D radiographs and CBCT, two references were taken into account: the length of the implant (8, 9 or 11 mm, depending on the case) and the inter-thread distance in the middle of the implant body (0.66 mm). The two measurements were averaged to obtain the final recorded figure. In instances where a minimum MBL was posited, the inter-microthread spacing of 0.22 mm was employed as a point of reference.

All measurements are expressed to one decimal place.

### 2.6. Statistical Analysis

The sample was first treated with descriptive analysis. Survival curves were compared using the Kaplan–Meier test and the log-rank test. Inter-observer reproducibility is assessed by intra-class correlation coefficient (ICC) and the mean of the 3 operators was calculated. A non-parametric approach was used with the Wilcoxon test to compare differences in mesial and distal MBL or vestibular and lingual MBL on CBCT. The Mann–Whitney test was used for distributions between two levels of any independent factor, and Spearman’s correlation was used to assess the degree of non-linearity between MBL and ordinal or continuous variables.

A Mann–Whitney test achieved 78% power to detect differences in the MBL distributions of two groups consistent with a large effect size (d = 0.8) at the 95% confidence level. The significance level used in the analyses was 5% (α = 0.05).

## 3. Results

Forty-eight sloped platform implants placed in the posterior mandible of 43 patients were included in the sample: 12 OsseoSpeed TX and 36 OsseoSpeed EV. There were 32 females (66.7%) and 16 males (33.3%), with a mean age of 51.9 ± 13.4 years at the time of surgery. A total of 81.2% received a single-unit prosthesis, and the remainder were included in a partial bridge. Patient demographics are shown in Table 1.

The mean time from surgery to the last periapical radiograph was 37.8 ± 30.9 months and to the last CBCT was 45.1 ± 37.2 months, both with a median of 29 months.

The overall survival rate was 93.8% (95% CI: 82.8–98.7%). There were 3 late failures (6.3%; 95% CI: 1.3–17.2%) occurring at a median of 43.7 months (range: 33–52 months), resulting in an estimated probability of survival of 83.8% at the maximum follow-up of 127 months (Table 2, Figure 4). Two of the failures occurred in smokers and the third in a patient taking strong psychoactive medication; all three presented early bone loss within the first six months after placement. During the observation period, no complications were recorded in the surviving implants, and all restorations remained functional without signs of peri-implant inflammation, maintaining patient satisfaction.

After applying log-rank tests to the independent factors recorded, no variable was associated with significantly different survival rates (Table 3). Regarding periodontal disease, no implant failures occurred in either group, and therefore the log-rank test could not be computed. This is indicated as NA in Table 3.

The intra-class correlation coefficients exceeded the threshold of 0.9, indicating excellent reproducibility (Table 4). Consequently, the mean of the three examiners was used for all MBL measurements.

The mean overall MBL (mesial and distal) from 2D periapical radiographs was 0.33 ± 0.61 mm, with a median of 0.05 mm (interquartile range [IQR]: 0–0.45 mm) (Table 5, Figure 5).

Although the distal MBL appeared higher than mesial, no significant differences were found (*p* = 0.053, Wilcoxon test). Regarding prevalence, 37.8% of the implants showed no detectable MBL (MBL = 0 mm), and 82.2% did not exceed 0.5 mm, which is considered a clinical success threshold (Figure 6).

Due to the large variation in follow-up time (6–127 months), the mean annual rate of MBL was also calculated, resulting in 0.19 ± 0.44 mm/year, with a median of 0.03 mm/year (IQR: 0–0.11).

In a subgroup of 22 patients, buccal and lingual MBL measurements were obtained by CBCT. The mean total MBL in this subset was 0.47 ± 0.66 mm, with a median of 0.16 mm (IQR: 0–0.67 mm). The distribution of buccal and lingual MBL values is presented in Figure 7. No significant differences were found between buccal and lingual measurements (*p* = 0.388, Wilcoxon test).

A weak but significant inverse correlation was observed between buccal MBL and implant length (r = −0.47, *p* = 0.028, Spearman). Moreover, buccal and total MBL were significantly higher in multiple-unit restorations compared to single-unit restorations (*p* = 0.001 and 0.003, respectively, Mann–Whitney test), as shown in Figure 8.

## 4. Discussion

The primary objective of the study was to address the survival of sloped platform Profile implants placed in the posterior mandible. The Cumulative Survival Rate (CSR) obtained was 93.7% in line with previous clinical studies [12]. However, a short prospective study of 238 Profile implants in the posterior mandible claimed a CSR of 99.5% with a mean follow-up of 28.7 months [9]. In the present study, there was a longer follow-up of 40.1 months, which could be responsible for the different CSR value, as the 2-year CSR in our study was still 100%.

In addition, a clinical study of the same mandibular oblique ridges treated with staged Guided Bone Regeneration (GBR) and regular geometry implants showed a CSR of 98.9% in one year [21]. Although the clinical results obtained were acceptable, it should be borne in mind that this was a very short-term result and that the treatment was more complex, prolonged (eight months of GBR healing) and involved at least two surgical procedures. Considering the patient’s perspective, GBR procedures seem to create more postoperative discomfort than conventional implant placement without GBR [22]. Indeed, a simplified treatment may be considered an advantage of the type of implant studied. Implant placement and restoration were considered easier, as concluded in a recent RCT comparing regular versus sloped implants [11]. There are two other options for surgically adapting the implant to a resorbed crest: placing it in a tilted position or using a narrow implant. Inclined implants mean choosing a position that is not ideal from a prosthetic point of view. The divergence between the axial axis of the implant and the prosthesis would require the latter to be cemented, which could increase the risk of biological complications [23], or to use a dynamic screw, which has shown promising results, but for which there is still a lack of research in the posterior region [24]. In the case of narrow implants in the posterior region, a higher mechanical risk has been found especially in diameters smaller than 3.25 mm and/or when not splinted with multiple implants [25].

The secondary outcome of this study was the assessment of MBL. In our study, MBL was calculated from prosthesis delivery to the most recent radiograph, which allowed us to distinguish early bone remodeling after loading from progressive marginal bone loss during follow-up. The mean MBL measured on two-dimensional radiographs was 0.27 and 0.39 mm mesially and distally, respectively, with an annual bone loss of 0.19 mm. These figures were comparable to those reported in other studies [10], thereby demonstrating the efficacy of this implant design in maintaining peri-implant bone levels.

Some authors consider MBL < 0.5 mm together with a follow-up time greater than 6 months as a representative criterion of treatment success [26], while others include this threshold to distinguish bone loss beyond the initial remodeling process [27]. Importantly, the 2017 World Workshop on Periodontology defined peri-implantitis as the combination of bleeding/suppuration on probing, probing depth ≥ 6 mm, and bone loss ≥ 3 mm [20]. Accordingly, in our study, MBL > 0.5 mm was not interpreted as peri-implantitis but rather as early remodeling or minimal changes within accepted success thresholds. In our sample, 82.2% of implants did not exceed an MBL of 0.5 mm after a mean follow-up of 44.2 months.

In a subset of patients, it was possible to have a CBCT prescribed for diverse reasons (i.e., a new implant treatment, etc.), and MBL was assessed buccally and lingually, obtaining mean values of 0.66 and 0.28 mm, respectively, and an annual mean MBL of 0.28 mm. Again, these figures were comparable to those reported in other similar investigations [28]. Despite the high inter-examiner agreement observed in the present study, several CBCT images were challenging to ascertain due to the absence of definition. Indeed, CBCT resolution is low compared to 2D radiographies, and the precision of its linear measurements are also highly dependent on the type of device and acquisition parameters, field of view, voxel size, kilovolts, etc. Furthermore, a recent systematic review has indicated that CBCT measurement may be discrepant for defects smaller than 1 mm. Especially, thinner bone walls are more susceptible to being obscured or misinterpreted due to artifacts [29].

In a noteworthy case, there was unanimous agreement between examiners regarding the mean MBL, which was recorded as 0 and 0.8 mm at the mesial and distal sites, and 4.5 and 0.1 mm, respectively, at the buccal and lingual sites. This discrepancy can be attributed to either a thin buccal plate with a post-remodeling dehiscence or the device’s resolution, given that the accuracy of buccal bone plate assessment on CBCT images has been considered to be significantly limited [30]. However, the beam hardening and scattering generated by metallic implants result in the formation of artifacts with severe streaking and shadowing, which impedes the precise delineation of bone edges in close proximity to the implant. To address this challenge, the utilization of deep learning software has been proposed as a potential solution [31].

Only in four implants placed in three patients, the MBL values exceeded 1.5 mm, representing less than 10% of the total sample. In one patient, flapless guided surgery was performed, and the two implants were left in a suboptimal vertical position. In this case, the relatively high MBL can be attributed to this rather than to biological remodeling of the marginal bone. In the other two patients, the bone loss is localized only distally and can only be associated with excessive initial bone remodeling. Nevertheless, both implants are functional and have a follow-up of 21 and 34 months.

As non-normal or asymmetric distribution, median figures could be more representative than the means. In this case, medians of 0.05 mm (interquartile range (IQR) 0–0.45 mm) obtained by 2D analysis and 0.16 mm (interquartile range: 0–0.67 mm) in the CBCT patient sample, were consistent with previous research [28].

Traditional management of oblique alveolar ridges relies on GBR techniques such as horizontal and vertical augmentation, particulate grafts, onlay blocks, or membrane-assisted regeneration. While effective, these procedures are technique-sensitive and associated with additional costs and patient morbidity. In our series, sloped implants allowed prosthetically driven placement in resorbed ridges, minimizing the need for GBR and simplifying treatment in selected cases.

Although no statistically significant associations were detected in the present sample, previous research, including our own prospective cohort [32], has demonstrated that bruxism and a history of periodontitis are associated with an increased risk of implant failure and with higher MBL values. In that study, bruxism significantly increased the risk of early failure, and periodontitis showed a clear tendency toward greater bone loss. The absence of significant results in our cohort may be explained by the smaller sample size and retrospective design, which limit the statistical power to detect such associations.

These findings emphasize the clinical value of sloped implants to simplify treatment planning in cases with moderate mandibular ridge resorption.

The retrospective design of this study inherently limits the control of confounding factors and introduces potential selection bias, as treatment and follow-up were not standardized prospectively.

In addition, keratinized gingiva width and soft tissue phenotype were not systematically recorded, which prevented analysis of their potential influence on marginal bone remodeling.

The relatively small sample size further reduced the statistical power to detect associations with independent variables.

Finally, the absence of data on clinical attachment and bleeding on probing impoverishes the description of implant evolution. Having these parameters, together with serial radiographs during follow-up, would have allowed a clearer distinction between progressive MBL and initial remodeling.

## 5. Conclusions

The survival rates and marginal bone loss (MBL) of profile implants are comparable to those of conventional flat shoulder implants. In cases of mild atrophic ridge with an oblique shape, sloped-shoulder implants may facilitate treatment and enhance patient comfort by reducing the need for GBR. Therefore, they may be a suitable option for restoring mandibular posterior areas exhibiting moderate resorption, offering potential advantages in specific clinical scenarios.

## Figures and Tables

**Figure 1 dentistry-13-00466-f001:**
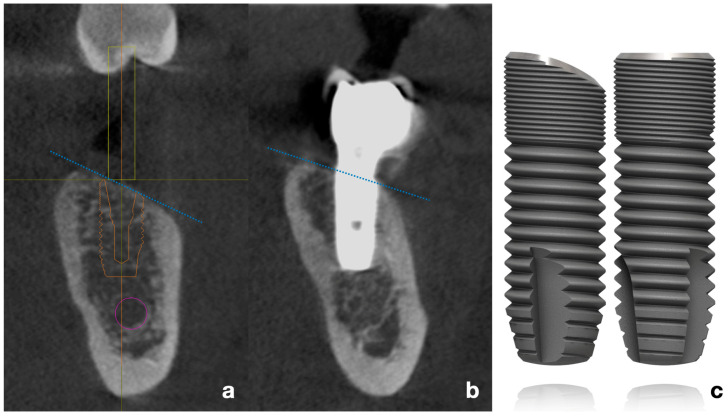
(**a**) Virtual planning of implant placement following the inclination of the alveolar crest. (**b**) CBCT bucco-lingual section of a representative case, showing the implant position relative to the mandibular ridge. Blue dotted lines represent the typical inclination of the resorbed posterior mandibular crest (**c**). Schematic drawing of flat-shoulder vs. sloped-shoulder implant (OsseoSpeed EV). /Copyright Dentsply Sirona Implants. Used with permission/.

**Figure 2 dentistry-13-00466-f002:**
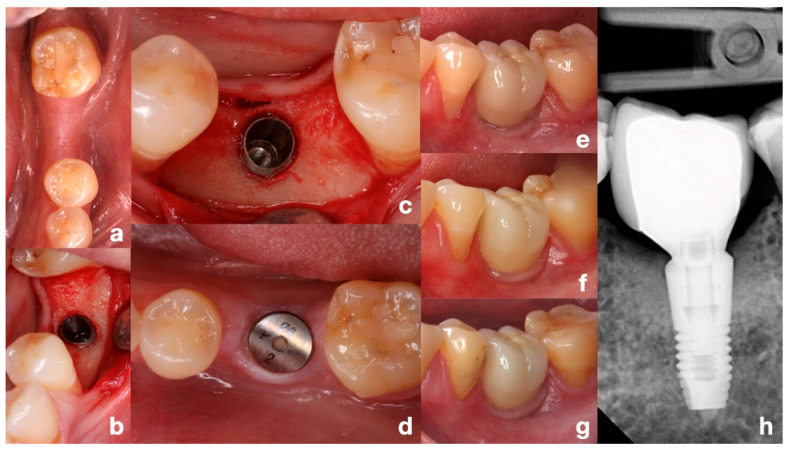
Representative clinical case with 10-year follow-up. (**a**) Initial situation of the edentulous posterior mandible, occlusal view. (**b**,**c**) Implant placement adapted to the angulated ridge, intraoperative direct and occlusal views. (**d**) Healing of soft tissue after non-submerged technique with healing abutment. (**e**) Clinical view immediately after prosthesis placement. (**f**,**g**) Clinical views at 5 and 10 years of function. (**h**) Periapical radiograph at 10 years showing stable bone levels. Note: This is an illustrative individual case. The median follow-up of the study population was 44.2 months.

**Figure 3 dentistry-13-00466-f003:**
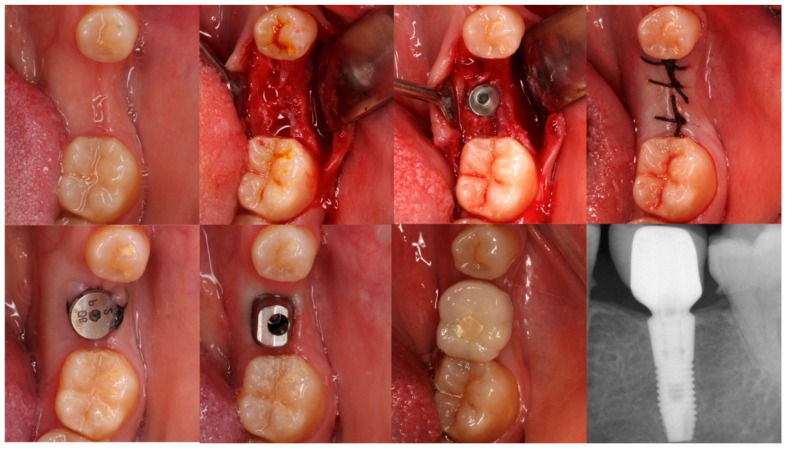
Sequential occlusal views of implant placement and restoration in the left mandibular first molar, using a sloped-shoulder implant. This is the only case in the series placed with a submerged technique, which was not considered in survival or MBL analyses due to being a single occurrence. The final periapical radiograph shows the 3-year follow-up outcome.

**Figure 4 dentistry-13-00466-f004:**
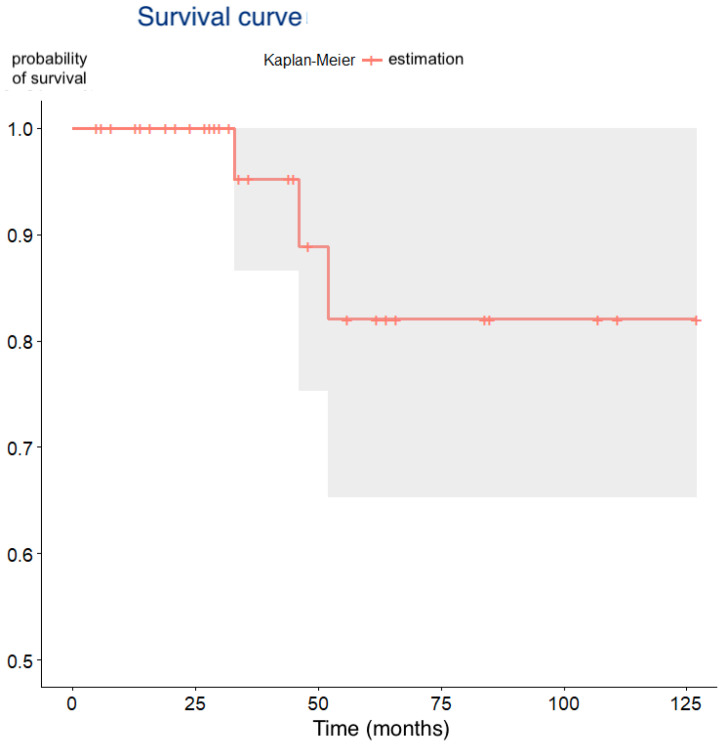
Kaplan–Meier survival curve showing cumulative implant survival over time (months).

**Figure 5 dentistry-13-00466-f005:**
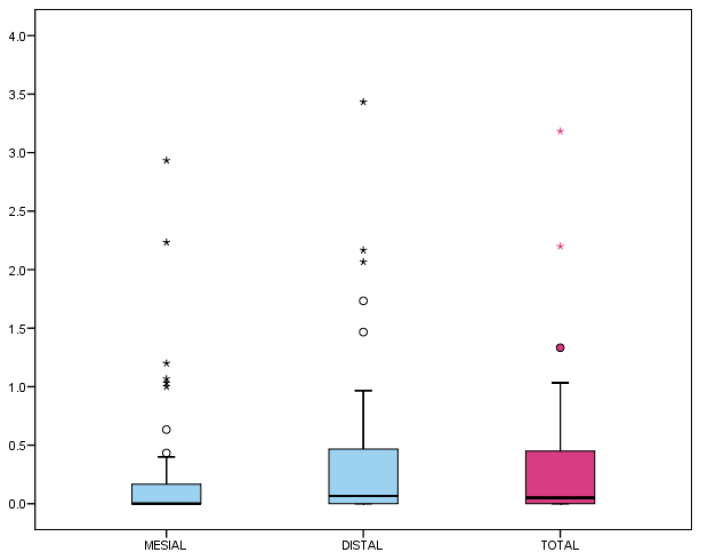
Marginal bone loss (MBL) from 2D periapical radiographs (mesial and distal). Box-plot representation: the box encloses 50% of values (interquartile range), with the median as the central line. Whiskers indicate the 5th–95th percentiles; outliers are represented as circles and extreme values as asterisks.

**Figure 6 dentistry-13-00466-f006:**
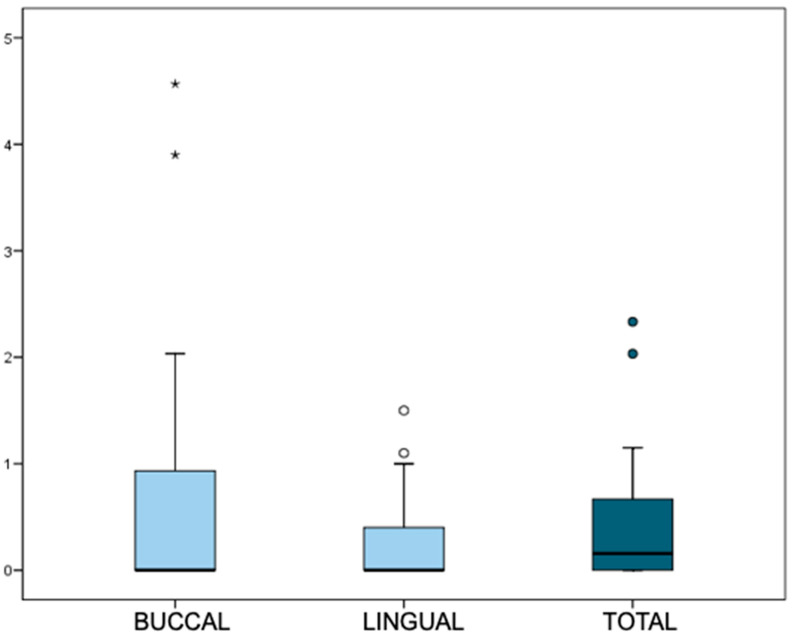
Prevalence of MBL >0 mm and > 0.5 mm at mesial, distal, and total sites. Box-plot representation: the box encloses 50% of values (interquartile range), with the median as the central line. Whiskers indicate the 5th–95th percentiles; outliers are represented as circles and extreme values as asterisks.

**Figure 7 dentistry-13-00466-f007:**
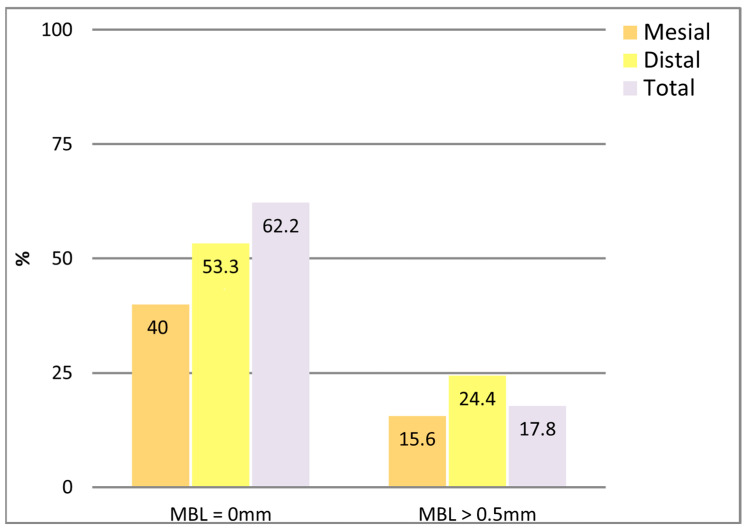
MBL values from 3D CBCT measurements (buccal and lingual).

**Figure 8 dentistry-13-00466-f008:**
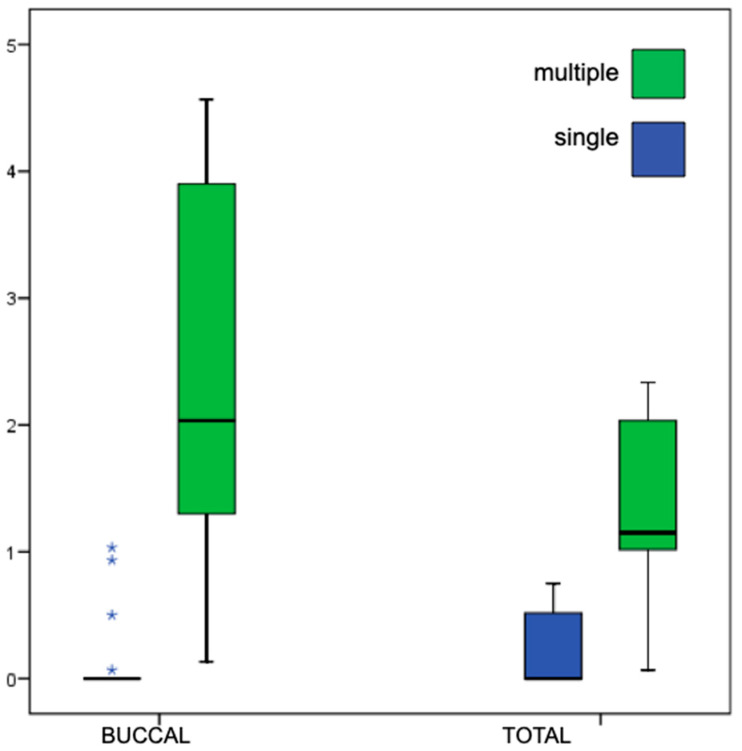
Comparison of MBL values from CBCT according to prosthesis type (single vs. multiple units). Box-plot representation as described in Figure 5, whiskers indicate the 5th–95th percentiles, and extreme values as asterisks.

**Table 1 dentistry-13-00466-t001:** Baseline patient and implant characteristics, including demographic variables, implant dimensions, site distribution, prosthesis type, abutment type, retention mode, and health conditions. Values presented as absolute numbers and percentages.

	Characteristic	*n*	%
Age (years)	51.9 (±13.4)		
Gender	Male	16	33.3
	Female	32	66.7
Profile Implant System	OsseoSpeed TX	12	25
	OsseoSpeed EV	33	75
Implant Position	Second Premolar	3	6.3
	First Molar	35	72.9
	Second Molar	10	20.8
Implant Diameter	4.2	31	64.6
	4.5	1	2.1
	4.8	5	10.4
	5	11	22.9
Implant Length	8	4	8.3
	9	20	41.7
	11	22	45.8
	13	2	4.2
Type of prosthesis	Single unit	39	81.2
	Multiple	9	18.8
Type of abutment	Customized	41	85.4
	Stock	7	14.6
Type of retention	Screwed	43	89.6
	Cemented	5	10.4
Health Conditions	Periodontal	19	42.2
	Smoking	15	31.3
	Bruxism	12	25

**Table 2 dentistry-13-00466-t002:** Kaplan–Meier survival estimates for implant failure over time, showing number at risk, failures, censored cases, interval survival, and cumulative survival rates.

Time (Months)	Implants at Risk	Failed Implants	Drop-Out/Lost to Follow-Up	Implant Survival in the Interval	Cumulative Survival
0–12	48	0	9	100%	100%
12–24	39	0	10	100%	100%
24–36	29	1	11	96.5%	96.5%
36–48	17	1	3	94.1%	90.8%
48–60	13	1	2	92.3%	83.8%
60–72	10	0	4	100%	83.8%
72–84	6	0	1	100%	83.8%
84–96	5	0	1	100%	83.8%
96–108	4	0	2	100%	83.8%
108–120	2	0	1	100%	83.8%
120–127	1	0	1	100%	83.8%

**Table 3 dentistry-13-00466-t003:** Distribution of implant failures by patient/implant factors. Results of log-rank test are presented. For periodontal disease, no failures occurred in either group; therefore, *p*-value is not applicable (NA).

	*p*-Value
Gender	0.844
Age	NA
Medical Antecedents (yes/no)	0.323
Smoking	0.122
Occlusal Splint	0.218
Periodontal Disease	NA
Implant System	0.193
Diameter (</>=4.5 mm)	0.313
Length (</>10 mm)	0.250
Tooth (PM/M1/M2)	0.250
Screw/Cement	0.353
Single/Multiple Prosthesis	0.398

**Table 4 dentistry-13-00466-t004:** Inter-examiner reproducibility of radiographic MBL measurements, expressed as intra-class correlation coefficient (ICC) with 95% confidence intervals.

	ICC	CI 95%
MBL RX Mesial	0.930	0.889–0.959
MBL RX Distal	0.942	0.908–0.966
MBL CBCT Vestibular	0.970	0.941–0.986
MBL CBCT Lingual	0.935	0.875–0.970

**Table 5 dentistry-13-00466-t005:** Descriptive statistics of 2D periapical MBL measurements (mesial, distal, total): mean, standard deviation, range, and percentiles (P25, median, P75).

	N	Mean	SD	Min	Max	P_25_	Median	P_75_
MBL Mesial	45	0.27	0.60	0.00	2.93	0.00	0.00	0.17
MBL Distal	45	0.39	0.72	0.00	3.43	0.00	0.07	0.47
MBL Total	45	0.33	0.61	0.00	3.18	0.00	0.05	0.45

## Data Availability

The data presented in this study are available on reasonable request from the corresponding author. The data are not publicly available due to privacy restrictions.

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
