# Peer review of "Clinical and Radiographic Outcomes of Sloped-Shoulder Implants in the Posterior Mandible: A Retrospective Study"

_dentistry, 2025, doi:10.3390/dj13100466_

Round 1
Reviewer 1 Report
Comments and Suggestions for Authors
The main objective of the study was to evaluate the survival of implants installed with various inclinations in the molar region of the mandible.
I find the article interesting. It clearly highlights the need for a correct diagnosis for good implantology treatment, considering the inclination of the dental implant placement. I believe this is true.
The major shortcoming of this article is that the methodology, and indeed the entire text, is confusingly described.
The English and figure descriptions also need to be improved so we can better review this paper.
I can't suggest how to improve the statistical analyses, but they also seem very confusing.
The bibliographic references could also be improved.
The English and figure descriptions also need to be improved so we can better review this paper.
There are several types of guided bone regeneration (GBR) that need to be included in this text more clearly.
"Initial clinical studies have shown that the implant's inclined shoulder facilitates precise placement, consistent with the planned prosthetic rehabilitation, as there is less need for positional adjustments to the oblique alveolar ridge." This statement in the text seems to me to clinically suggest better implant anchorage in cortical bone and needs to be further discussed or evidenced in the text.
Comments on the Quality of English LanguageThe English and figure descriptions also need to be improved so we can better review this paper.
Author Response
Comment 1: The methodology and text are confusingly described.
Response: The Materials and Methods section was reorganized to clearly detail the retrospective design, inclusion criteria, variables, and ethics approval (Hospital of Elche, PI 106/2023). The Committee waived informed consent due to the retrospective and anonymized nature of the data. This information was also summarized in the Abstract.
Comment 2: The English and figure descriptions need to be improved.
Response: The entire manuscript was revised by a fluent English speaker. All figure legends were rewritten to be self-explanatory and consistent.
Comment 3: Several types of GBR need to be included more clearly.
Response: The Introduction and Discussion were expanded to describe horizontal and vertical GBR, particulate grafts, onlay blocks, and membrane-assisted regeneration, and to discuss how sloped implants may reduce their need.
Comment 4: The statement suggesting “better anchorage in cortical bone” should be clarified.
Response: The sentence was rephrased to: “The inclined shoulder of the implant facilitates prosthetically driven positioning by adapting to the oblique alveolar ridge, thereby reducing the need for GBR or compensatory adjustments.”
Reviewer 2 Report
Comments and Suggestions for Authors
hank you for your submission. The study presents valuable insights; however, several points require clarification to ensure compliance with ethical standards and improve data transparency:
1.Table 3 – Formatting and Completeness
The results are generally well-presented; however, Table 3 contains a missing p-value for the variable “Periodontal Disease.” Please complete this entry or remove the row if the data were not analyzed.
2.Study Duration and Ethical Approval
The manuscript reports a retrospective study spanning over a decade, yet the ethics approval was granted only in December 2023. The author should justify a. How patient consent was managed throughout this extended timeframe. b. How was long-term follow-up data ethically justified and documented, particularly for cases included in CBCT analysis.
The English could be improved to more clearly express the research.
Author Response
Comment 1: Table 3 has a missing p-value for “Periodontal Disease.”
Response: We clarified that no implant failures occurred in either group; therefore, the log-rank test could not be computed. The entry now appears as NA in Table 3 and is explained in the Results.
Comment 2: Ethics approval was granted in 2023 although the study spans a decade. Clarify consent and justification.
Response: Clarified in Methods and Abstract: data were retrieved from routine records, the CEIm approved the retrospective study, and informed consent was explicitly waived due to anonymization and absence of additional risk.
Comment 3: English quality should be improved.
Response: The manuscript has been carefully revised for clarity and readability.
Reviewer 3 Report
Comments and Suggestions for Authors
This retrospective study aims to evaluate the survival rate and radiographic marginal bone loss (MBL) of sloped platform implants in the posterior mandible, and to identify factors predictive of these outcomes. While the objective of the manuscript is clear, the report leaves several key questions unanswered:
- The limitations and potential biases inherent in a retrospective study design should be discussed.
- The timing of MBL measurements is unclear. The authors mention bone remodeling following implant loading—please clarify and discuss the factors influencing both MBL and bone remodeling.
- Please provide a definition of peri-implantitis. According to data from this study, 17.8% of implants had MBL > 0.5 mm (World Workshop, 2017). Is this prevalence of peri-implantitis unusually high for a study with a 40-month follow-up?
- Could the high prevalence of patients with a history of periodontitis, bruxism, or smoking have influenced the MBL outcomes? Please elaborate and provide an analysis.
- The authors state that all implants were treated using a non-submerged technique (platform 4). However, Figure 3 presents a case involving a submerged technique. Could the surgical approach (submerged vs. non-submerged) affect treatment outcomes such as MBL, especially in patients with a thin tissue phenotype? Please elaborate and analyze.
- Figure 2 shows an implant with a narrow band of keratinized gingiva (KG). Did the authors evaluate KG width and gingival phenotype in relation to MBL outcomes?
Given the concerns outlined above, the reviewer recommends major revision of the manuscript.
Author Response
Comment 1: Discuss limitations and biases of retrospective design.
Response: A Limitations section was added: the retrospective design introduces potential selection bias, lacks prospective standardization, and reduces control of confounders. This limitation is also summarized in the Abstract.
Comment 2: Timing of MBL measurements is unclear.
Response: Clarified in Methods, Discussion, and Abstract: MBL was calculated from prosthesis delivery (baseline) to the most recent radiograph, allowing distinction between early remodeling and long-term loss.
Comment 3: Define peri-implantitis and clarify MBL >0.5 mm.
Response: The 2017 World Workshop definition was added in Methods and Discussion. We emphasized that MBL >0.5 mm was not interpreted as peri-implantitis but as minimal remodeling. This clarification was also summarized in the Abstract.
Comment 4: Could periodontitis, bruxism, or smoking influence MBL?
Response: Discussion expanded: although no significant associations appeared in this cohort, previous evidence (including our JOMI 2023 prospective study) shows bruxism and periodontitis increase failure risk and MBL. Lack of significance here may be due to sample size and retrospective design.
Comment 5: Figure 3 shows a submerged technique, but text states all implants were non-submerged.
Response: Clarified in Clinical procedure: all implants were placed non-submerged except for one intentionally submerged case (Figure 3), which was excluded from survival/MBL analyses. The figure legend was updated.
Comment 6: Was keratinized gingiva or gingival phenotype evaluated?
Response: These variables were not systematically recorded. This was added as a limitation in the Discussion.
Round 2
Reviewer 1 Report
Comments and Suggestions for Authors
Congrats!
I only suggest a schematic drawing showing the various types of dental implant inclination with emphasis on the tilted implant as suggested in this article.
Author Response
We sincerely thank the reviewer for the valuable suggestion. Following this recommendation, we have modified Figure 1 by adding a schematic drawing (Figure 1c) to illustrate the concept more clearly. We believe this addition improves the clarity of the manuscript.
Reviewer 3 Report
Comments and Suggestions for Authors
well revised
Author Response
We sincerely thank the reviewer for the positive feedback. We are pleased that the revisions were found satisfactory. No additional changes were required beyond those already implemented.